# Audio-Driven Identity Manipulation for Face Inpainting

## ABSTRACT

Recent advances in multimodal artificial intelligence have greatly improved the integration of vision-language-audio cues to enrich the content creation process. Inspired by these developments, in this paper, we first integrate audio into the face inpainting task to facilitate identity manipulation. Our main insight is that a person's voice carries distinct identity markers, such as age and gender, which provide an essential supplement for identity-aware face inpainting. By extracting identity information from audio as guidance, our method can naturally support tasks of identity preservation and identity swapping in face inpainting. Specifically, we introduce a dual-stream network architecture comprising a face branch and an audio branch. The face branch is tasked with extracting deterministic information from the visible parts of the input masked face, while the audio branch is designed to capture heuristic identity priors from the speaker's voice. The identity codes from two streams are integrated using a multi-layer perceptron (MLP) to create a virtual unified identity embedding that represennts comprehensive identity features. In addition, to explicitly exploit the information from audio, we introduce an audio-face generator to generate an 'fake' audio face directly from audio and fuse the multi-scale intermediate features from the audio-face generator into face inpainting network through an audio-visual feature fusion (AVFF) module. Extensive experiments demonstrate the positive impact of extracting identity information from audio on face inpainting task, especially in identity preservation.

## CCS CONCEPTS

• **Human-centered computing** → **Interaction design**; • **Computing methodologies** → **Image and video acquisition**; *Animation*; *Image manipulation*; **Computer vision tasks**.

## KEYWORDS

face inpainting, audio, multi-modal

## 1 INTRODUCTION

Audio and image are the two most common modes that people use to perceive the world[10]. They are closely related, especially when it comes to recognizing other people or sensing their emotional states[35]. People are often able to build a mental model of what a person looks like based on a voice, because vocal and visual signals are highly correlated[19, 42]. A large number of studies have proved that a person's biophysical parameters, such as gender, age,

Permission to make digital or hard copies of all or part of this work for personal or classroom use is granted without fee provided that copies are not made or distributed for profit or commercial advantage and that copies bear this notice and the full citation on the first page. Copyrights for components of this work owned by others than the author(s) must be honored. Abstracting with credit is permitted. To copy otherwise, or republish, to post on servers or to redistribute to lists, requires prior specific permission and/or a fee. Request permissions from permissions@acm.org.
*ACM MM, 2024, Melbourne, Australia*
© 2024 Copyright held by the owner/author(s). Publication rights licensed to ACM.
ACM ISBN 978-x-xxxx-xxxx-x/YY/MM
https://doi.org/10.1145/nnnnnnn.nnnnnnn

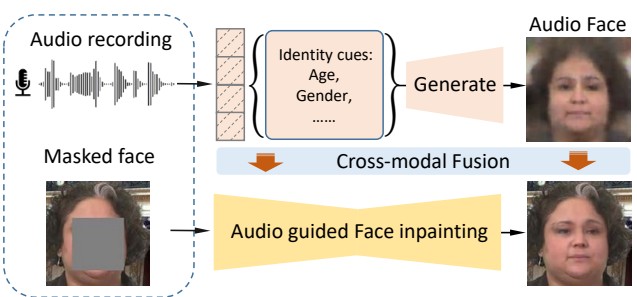

**Figure 1: Illustration of our pipeline. We extract a high-dimensional implicit identity embedding from audio and generate an audio face to fully exploit the identity information in audio. This information contains identity clues, such as age and gender, which can serve as guidance for the face inpainting network.**

health status, *etc*, can be inferred from the voice[32, 37]. What's more, by modeling the relationship between audio and image, researchers have drawn faces directly from speech clips[35, 47]. Although the images were not clear enough, they proved to be closely related to a person's identity.

There is a relationship between a person's voice and his facial structure[47]. On the one hand, the underlying skeletal and articulator structure of the face directly affects the shape, size and other acoustic characteristics of the vocal tract, resulting in different voices that people make[33, 41]. On the other hand, objective factors such as gender, age, environment, *etc*, are all important components of voice characteristics[21]. In fact, it turns out that each of these factors can be inferred independently from voices or faces[22, 23]. Thus, by associating key facial attributes with important features of voices, the audio information may assist in generating a more restored face when removing face occlusion or repairing faces.

Face inpainting is an important work, which is widely used in many fields, such as blemish restoration[1], occlusion removal[5, 24], face makeup[9], blink repair[8, 49], *etc*. In recent years, deep-learning based face inpainting methods have achieved good results[11, 30, 44]. Thanks to the widespread use of GAN[14] and Transformer[45], they can generate realistic looking faces without maintaining the face identity. Different from natural images, faces are more structured and have their own unique identity characteristics. Face inpainting tasks can be oriented to more practical needs and complement other face tasks, such as face recognition[13], face detection[17], facial attribute classification[7], *etc*, which have high requirements for the preservation of face identity.

In order to preserve or complete the face identity, some methods try to introduce external prior knowledge. [18, 52] constraint identity consistency by calculating semantic feature losses on the output and ground truth, which is more like feature regularization than completion. Some other methods [8, 25, 40] assume that

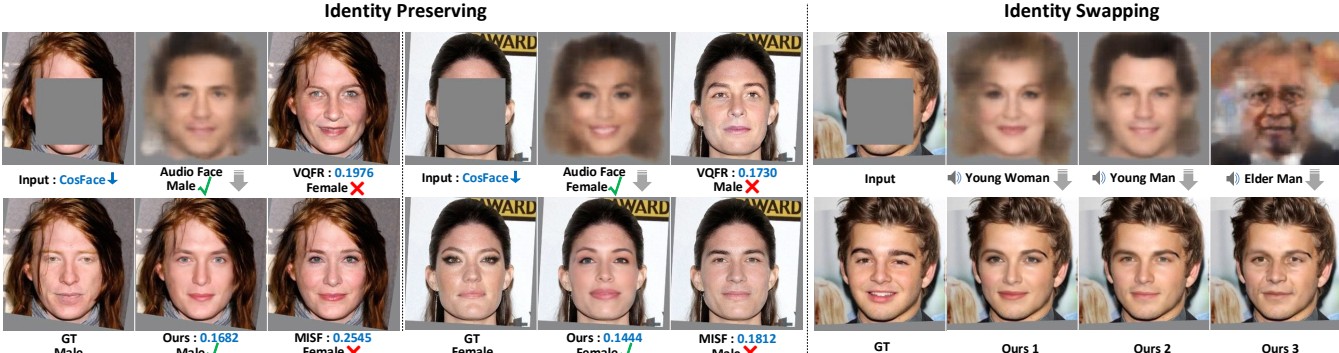

**Figure 2: Visual results of our methods in identity preserving and identity swapping. When the input audio and face are from the same person, our method uses the reference face generated from audio to achieve better identity preservation than recent methods. They mistakenly generate a male face for the input female face or a female face for the input male face. We use a face recognition network CosFace [46] to measure the identity distance, which is lower for better. On the other hand, when the input audio and face are from different people, our method can be used to perform an audio-control identity swapping. Given an audio of a young woman, a young man or an elder man, our method can generate a face with corresponding features.**

reference images are available when making inferences. They extract key identity attributes, such as eyes, from reference images to compensate for the loss of identity. [28] design a cross-modal disentangling network to extract identity information both from text and reference images for eyeglasses removal. While these methods demonstrate the advantages of reference images and text for identity preservation, these references are not readily available in practical applications such as surveillance analysis. With the development of multimedia, it is easy to obtain a face that is speaking from videos, which may be covered by a mask and need to be repaired. Therefore, we propose an audio-driven face inpainting approach, which infers face identity from audio to achieve high-fidelity face generation.

In general, our method is a dual-stream network including a face branch and an audio branch, as shown in Figure 1. In the face branch, we encode the masked faces into a high-dimensional face embedding representing deterministic identity prior derived from the unmasked regions. In the audio branch, we use a pre-trained audio embedding network to extract the audio identity embedding, which is a heuristic prior that implicitly expresses identity information about the face. Then, we fuse the two codes and obtain a complete identity code through an MLP. We also introduce an identity embedding loss for constraining the complete identity features with the face identity labels. Although our method can integrate the identity information of the audio in this way, it is difficult for a single face decoder to decode this implicit representation. Therefore, we introduce an additional audio face decoder to reconstruct faces from audio identity embedding, through which we pass the intermediate multi-scale feature maps to the face decoder as low-level semantic complements. We fuse features from two decoder with an audio-visual feature fusion (AVFF) module and generate a final face. In the end, we apply an identity consistency loss to constraint the final face and the audio face. We show some identity preservation and swapping results in Figure 2.

In general, this paper has the following main contributions:

- For the first time, we introduce audio into face inpainting for face identity preservation and leverage implicit representation and explicit features for identity reasoning.
- We design an audio-visual feature fusion (AVFF) module to fuse multi-scale features from the face and audio decoder. He learns an attention map containing global and local information for better feature fusion.
- We introduce an identity embedding loss and an identity consistency loss. Identity embedding loss is used to generate completed identity features, and identity consistency loss is used to constrain the feature consistency between the final face and audio face.
- We pre-process the previous audio-face dataset to obtain a high-quality audio-face paired dataset and demonstrate that our method performs better in generating high-fidelity faces than state-of-the-art methods.

## 2 RELATED WORK

**Face Inpainting.** Image inpainting aims to reconstruct the missing areas of the input images. Most of the existing image inpainting methods [11, 44] reasonably infer the missing pixel through the information around the hole. Compared with natural images, face images have stronger topological structure and local coherence. Therefore, it is of great significance to effectively predict the structure of faces by using the information around the hole. CSA [29] proposes a coherent semantic attention layer that better retains the missing structural information of the images. By recurrent feature reasoning, RFR [26] continuously fuses reasonable pixels around the hole to produce clear results. MISF [27] focuses on the smoothness between adjacent pixels of the images, which can realize high-fidelity image restoration. ICT [44] introduces transformer into the image inpainting task for the first time, which is used to reconstruct the structural priors of the images. VQFR [15] realizes high-quality blind face restoration based on the vector quantization dictionary and parallel decoders.

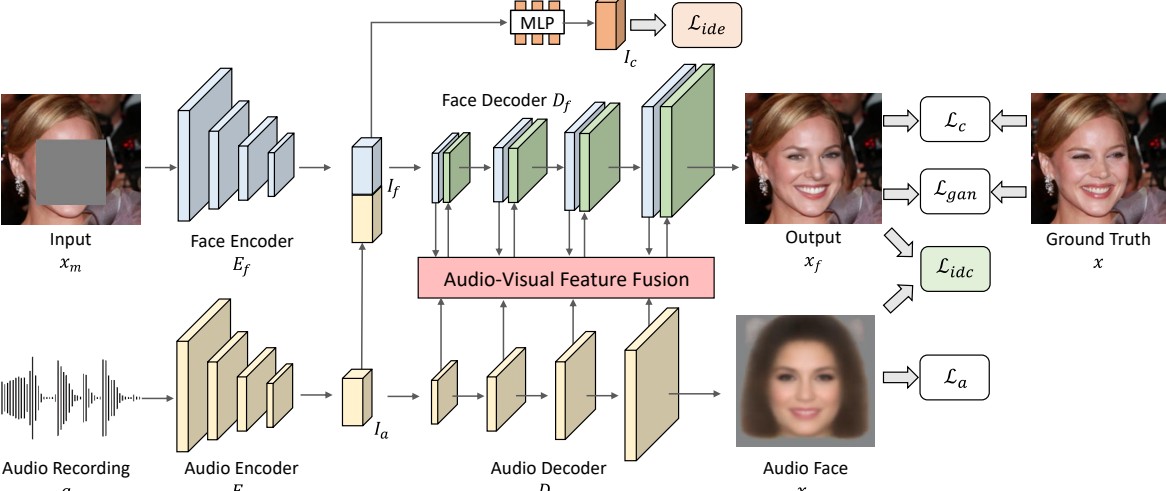

**Figure 3: Pipeline Overview. Our methods is a dual-stream network with a face branch and an audio branch. In the face branch, we extract identity features from face and audio, respectively, which is described as $I_i$ and $I_a$. Then, we concat these two implicit codes and obtain one completed face identity code $I_c$ through an MLP. After that, we fuse multi-scale features from two decoders in the proposed Audio-Visual Feature Fusion module and sent to the coarse face decoder to generate a coarse face.**

Although these methods try to reconstruct the face structures, they do not consider the identity knowledge of the faces. A person's voice has a strong correlation with his face structure. Thus, the voice may help restore the topology of faces, and the rich face identity information contained in voices may be conducive to maintaining the original attributes of faces.

**Identity from audio** Inferring speaker identity from audio is a long-standing task. Early work [20, 38] designed some hand-crafted features to map audio into a compact low-dimensional identity space for speaker identification. In recent years, some methods [4, 34] extend the representation to a much higher dimension to adequately extract speaker-discriminating features by deep learning network. More explicitly, some methods [12, 16, 50] predict specific identity attributes, such as age, gender, etc., directly from audio. These methods demonstrate that audio can provide a rich identity information supplement for face inpainting.

**Face reconstruction from audio** Reconstructing faces from audio has received much attention in recent years. A few methods directly learn an audio-face mapping from large data without any face prior. Y Wen [48] et al. use GAN to train a face generator, using one discriminator to determine real or fake faces and another to discriminate identities. Similarly, Speech2Face [35] trained a highly capable decoder on a million spectra-face data pairs to generate audio-visual identity-consistent faces. Nevertheless, these two methods can only generate relatively low-quality faces. On the other hand, talking face [3, 53, 54] has become a hot topic of recent research. It aims at generating mouth-synchronized faces from audio, so it focuses more on the content of the audio than on the identity. In contrast, our method aims to extract the speakers' identity and ignores the audio's content.

## 3 APPROACH

Given a person's masked face and an audio recording, our method aims to infer identity characters from audio and generate an identity-preserving face. The overview of our method is shown in Figure 3. We masked a face $x \in \mathbb{R}^{H \times W \times 3}$ with a large rectangle mask $\mathbf{m} \in \{0, 1\}^{H \times W \times 1}$ indicating the pixels need to be inpainted (with value 1) or not (with value 0). The audio recording is processed into a log Mel spectrogram with a fixed size. The masked face and the log mel-spectrogram are denoted as $x_m$ and $a \in \mathbb{R}^{h \times w}$.

Our method includes a face branch and an audio branch. In the face branch, we send $x_m$ into a face identity encoder to extract an identity embedding $I_f$ from the remaining area of the input face. Similarly, in the audio branch, an audio identity embedding $I_a$ is generated from audio using a pre-trained speaker recognition network. $I_f$ and $I_a$ are first concatenated to generate a completed face identity embedding $I_c$, then fed into different decoders reconstructing a inpainted face $x_f$ and an audio face $x_a$. In addition to implicit identity embedding fusion, multi-scale features from two decoders are reasonably fused in an audio-visual feature fusion (AVFF) module. We describe the fusion process in Section 3.1. We also introduce an identity embedding loss and an identity consistency loss (described in Section 3.2 ) to constrain the completed identity embedding and the consistency between the final face $x_f$ and audio face $x_a$.

## 3.1 Audio-Visual Identity Fusion

**Identity Embedding Fusion.** The face identity embedding $I_f$ from the face branch can be considered a deterministic prior learned from the visible areas of the face. In contrast, the audio identity

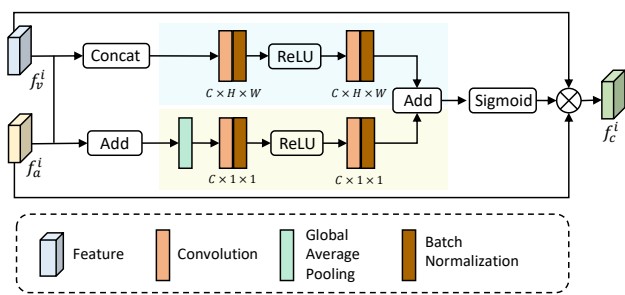

**Figure 4: Audio-Visual Feature Fusion Module. Since the input features $f_v^i$ and $f_a^i$ are misaligned, we fuse them with an attention map generated from a local and global branch. We concatenate two features in the local branch and extract local information with several convolution layers, while the global branch adds two features and uses a global average pooling to extract global information. Finally, we fuse them into an integrated feature map $f_c^i$.**

embedding $I_a$ from the audio branch is a heuristic prior to indicating implicit facial identity descriptions such as age, gender, etc. As $I_f$ is uncompleted, we take $I_a$ as an additional inference cue to achieve identity completion. As both $I_f \in \mathbb{R}^C$ and $I_a \in \mathbb{R}^{C'}$ ($C = 512, C' = 64$) are 1D vectors, we directly concatenate them along channel dimension and pass them through a linear layer to generate an intermediate feature $f \in \mathbb{R}^{C \times 1 \times 1}$ sending into the face decoder. In addition, we take an MLP as a modulator mapping $f$ into a high-dimensional space (set to 2048 by default ) represented as the completed face identity $I_c$.

**Audio Face Decoder.** Although the intermediate features $f$ include identity information from both face and audio, it is difficult to reconstruct an audio-visual consistent face from a single face decoder $D_f$. An immediate consequence is no difference in the generated faces when the audio changes. To address this problem, we introduce an audio face decoder $D_a$ generating face directly from audio identity embedding $I_a$. The middle features in this decoder explicitly describe audio identity at the pixel level.

However, reconstructing faces from audio is not easy work because the network needs to learn the mapping relationship between audio and faces from a large amount of data. Although early work [48] has been attempted, the faces they generate are at a very low resolution ($64 \times 64$), making them unsuitable for generating high-quality faces. Instead of applying their method directly, we retrain it in a collected high-quality audio-face dataset. We also add several upsampling layers to fit a high resolution ($256 \times 256$). In the end, our audio face decoder takes audio identity embedding $I_a$ as input to generate an audio face $x_a \in \mathbb{R}^{H \times W \times 3}$.

**Audio-Visual Feature Fusion.** The feature from audio face decoder $D_a$ explicitly describe the identity information from audio. We extract mutli-scale feature vectors $f_v^i, f_a^i \in \mathbb{R}^{\frac{H}{r} \times \frac{W}{r} \times C_i}$ ($r = 2^i$ and $i \in \{1, 2, 3, 4\}$) from the face decoder $D_f$ and the audio face decoder $D_a$. Due to the pixel misalignment between input face and audio face, it is not reasonable to concatenate two features directly. In constrast, we propose an audio-visual feature fusion

| Datasets | VoxCeleb-ID Train | VoxCeleb-ID Test | FaceForensics++ Train | FaceForensics++ Test | HDTF Train | HDTF Test | Total |
|---|---|---|---|---|---|---|---|
| Identities | 763 | 190 | 611 | 150 | 287 | 71 | 2,072 |
| Faces | 5,697 | 1,383 | 6,110 | 1,500 | 2,870 | 710 | 18,270 |
| Standard Faces | 5,697 | 1,383 | 1107 | 231 | 556 | 143 | 9114 |
| Audio Segments | 14,364 | 3,563 | 1,814 | 429 | 9,992 | 2,253 | 32,415 |

**Table 1: Three pre-processed audio-face paired datasets. Standard Faces mean faces without lip movement.**

(AVFF) module to integrate two face features, shown in Figure 4. For the i th level, given two feature vectors $f_v^i$ and $f_a^i$, AVFF extract local and global information through two branches. The fusion process can be described as:

$$\widetilde{m} = \text{Sigmoid}(\xi_l(cat(f_v^i, f_a^i)) + \xi_g(G(f_v^i + f_a^i))),$$
$$f_r^i = \widetilde{m} * f_v^i + (1 - \widetilde{m}) * f_a^i. \tag{1}$$

where $cat$ denotes concatenate operation, $G$ denotes global average pooling. $\xi_l(\cdot)$ and $\xi_g(\cdot)$ indicate convolution, batch normalization, and ReLU in the local and global branches. The add operation is proved to be better than the concatenate operation in the global branch. $\widetilde{m}$ is an attention map representing the region of the two features focus on. After this process, we obtain an integrated feature graph $f_c^i$, which is subsequently sent to the face decoder and enter the next level.

## 3.2 Loss Functions.

Given a masked face $x_m$ and a audio recording $a$, our method can generate an final face $x_f$ and an audio face $x_a$, which are identity-consistent. The ground truth face and mask is denoted as $x$ and $\mathbf{m}$. We train our method with the following losses.

**Identity Embedding Loss.** In the process of identity embedding fusion, we integrate the face identity embedding $I_f$ from the face encoder and the audio identity embedding from the audio encoder and generate an identity embedding $I_c$ through an MLP. $I_c$ includes a definitive description of the unmasked face and inference of the identity attributes in audio so that it can be considered a complete identity description. In order to constrain this embedding consistent with the identity of ground truth face $x$, we extract its identity using a face identification network $\psi$ pre-trained on VGGFace2 [2] and calculate an L1 distance. We explain the identity embedding loss as follows:

$$\mathcal{L}_{ide} = \|I_c - \psi(x)\|_1 \tag{2}$$

**Identity Consistency Loss.** In addition to supervising that the predicted identity embedding and the real embedding are consistent, we find it necessary to supervise the identity consistency of the final face and the audio face, described in the following:

$$\mathcal{L}_{idc} = \|\psi(x_f) - \psi(x_a)\|_1 \tag{3}$$

In this way, we explicitly constrain that the final face refers to the identity properties of the audio face during the generation.

**Reconstruction Loss.** To maintain the similarity between the generated face and the real face, we calculate a L1 distance between the ground truth face and the final face, audio face respectively.

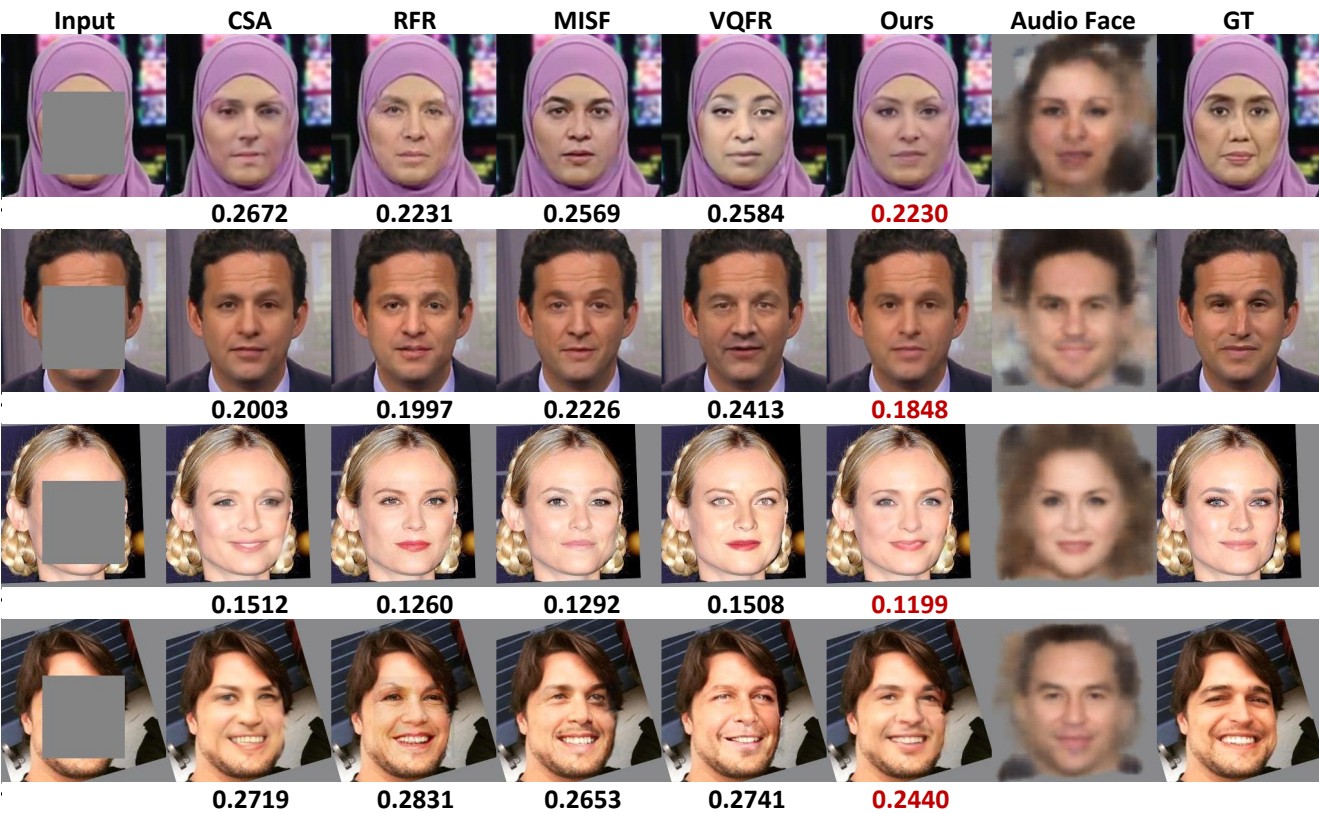

| | Input | CSA | RFR | MISF | VQFR | Ours | Audio Face | GT |
|---|---|---|---|---|---|---|---|---|
| | | 0.2672 | 0.2231 | 0.2569 | 0.2584 | 0.2230 | | |
| | | 0.2003 | 0.1997 | 0.2226 | 0.2413 | 0.1848 | | |
| | | 0.1512 | 0.1260 | 0.1292 | 0.1508 | 0.1199 | | |
| | | 0.2719 | 0.2831 | 0.2653 | 0.2741 | 0.2440 | | |

Figure 5: Qualitative results in three datasets. The image in the first row is from FaceForensics++, the second row is from HDTF, and the last two rows are from VoxCeleb-ID. We report the CosFace[46] distance blow the image. Previous methods may incorrectly inpaint a female face as a male, while our method can generate results that are most consistent with the ground truth face.

$$\mathcal{L}_c = \|x_f - x\|_1, \mathcal{L}_a = \|x_a - x\|_1 \qquad (4)$$

To be noticed, $\mathcal{L}_a$ can be considered as a regularization for audio face, and does not destroy the ability of pre-trained audio decoder to reason about the identity of the audio.

**GAN Loss.** Follow [29], we also introduce a GAN loss in to make the final image look more realistic. It is defined as:

$$\mathcal{L}_{gan} = \mathbb{E}[log(1 - D_w(x_f))] + \mathbb{E}[logD_w(x)] \qquad (5)$$

where D is the discriminator parameterized by $w$. We optimize our method with the global loss function:

$$\mathcal{L} = \lambda_{ide}\mathcal{L}_{ide} + \lambda_{idc}\mathcal{L}_{idc} + \lambda_c\mathcal{L}_c + \lambda_a\mathcal{L}_a + \lambda_{gan}\mathcal{L}_{gan} \qquad (6)$$

We set the loss weights as $\lambda_{ie} = 0.001$, $\lambda_{av} = 1$, $\lambda_c = 1$, $\lambda_a = 0.01$, $\lambda_{gan} = 0.002$.

## 4 EXPERIMENTS

### 4.1 Experimental Settings

**Data Preparation.** Faces in Previous audio-face datasets like VoxCeleb[34] are low resolution and blurred, which are unsuitable for generating high-quality faces. The face quality in the Celeb-ID [8] dataset

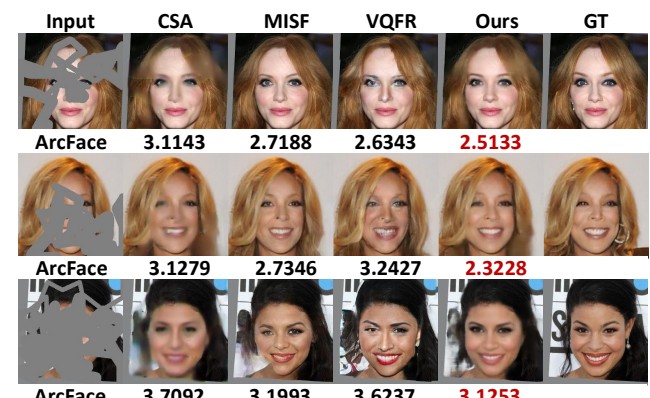

| Input | CSA | MISF | VQFR | Ours | GT |
|---|---|---|---|---|---|
| ArcFace | 3.1143 | 2.7188 | 2.6343 | 2.5133 | |
| ArcFace | 3.1279 | 2.7346 | 3.2427 | 2.3228 | |
| ArcFace | 3.7092 | 3.1993 | 3.6237 | 3.1253 | |

Figure 6: Qualitative results under three different sizes of irregular masks.

is much higher (with a resolution of 300x300), but it lacks audio recordings. Fortunately, both two datasets share some same identity labels. Therefore, to obtain high-quality face-audio pairs, we match faces in the Celeb-ID dataset and the audio recordings in

| Methods | FaceForensics++ | | | | | HDTF | | | | | VoxCeleb-ID | | | | |
|---------|----------|----------|----------|--------|-----------|----------|----------|----------|--------|-----------|----------|----------|----------|--------|-----------|
| | ↑ PSNR | ↑ SSIM | ↓ LPIPS | ↓ L1 | ↓ Landmark | ↑ PSNR | ↑ SSIM | ↓ LPIPS | ↓ L1 | ↓ Landmark | ↑ PSNR | ↑ SSIM | ↓ LPIPS | ↓ L1 | ↓ Landmark |
| CSA [29] | 28.1927 | 0.9154 | 0.0469 | 3.4886 | 5.9768 | 28.4556 | 0.9095 | 0.0522 | 3.5058 | 5.6826 | 25.8819 | 0.8947 | 0.0556 | 4.5279 | 8.7129 |
| RFR [26] | 25.8501 | 0.8957 | 0.0502 | 5.7996 | 6.1140 | 26.4366 | 0.8908 | 0.0530 | 5.5917 | 5.8179 | 25.3452 | 0.8879 | **0.0475** | 4.8986 | 8.7881 |
| ICT [44] | 23.6068 | 0.8379 | 0.0864 | 8.6667 | 13.1200 | 24.0702 | 0.8343 | 0.0780 | 8.2305 | 11.0897 | 22.6278 | 0.8436 | 0.0709 | 8.9450 | 12.6871 |
| MISF [27] | 27.8053 | 0.9120 | 0.0494 | 4.8085 | **5.7474** | 27.6142 | 0.8967 | 0.0540 | 5.1096 | 6.2175 | 25.3178 | 0.8873 | 0.0534 | 6.6094 | **8.4738** |
| VQFR [15] | 26.6317 | 0.8932 | 0.0535 | 5.3938 | 6.2453 | 26.4841 | 0.8767 | 0.0610 | 5.7760 | 6.2454 | 24.0869 | 0.8617 | 0.0598 | 7.4186 | 9.7937 |
| Ours | **28.2354** | **0.9166** | **0.0463** | **3.4811** | 5.9931 | **28.7509** | **0.9156** | **0.0514** | **3.3944** | **5.3709** | **25.9633** | **0.8963** | 0.0544 | **4.4863** | 8.7056 |

Table 2: Quantitative comparison with other methods in three datasets. We show best results in bold. Our method outperforms other methods in most metrics.

| Methods | FaceForensics++ | | | | | HDTF | | | | | VoxCeleb-ID | | | | |
|---------|-----|---------|-----------|--------|--------|-----|---------|-----------|--------|--------|-----|---------|-----------|--------|--------|
| | HOG | VGGFace | SphereFace | CosFace | ArcFace | HOG | VGGFace | SphereFace | CosFace | ArcFace | HOG | VGGFace | SphereFace | CosFace | ArcFace |
| CSA [29] | 2.1427 | 3.3881 | 0.3400 | 0.2318 | 3.8964 | 2.1607 | 3.1305 | 0.3574 | 0.2291 | 3.9095 | 2.1198 | 3.7891 | 0.3808 | 0.2168 | 3.6800 |
| RFR [26] | 2.2879 | 3.4696 | 0.4088 | 0.2434 | 3.9519 | 2.2763 | 3.1413 | 0.4154 | 0.2356 | 3.8526 | **2.0187** | 3.7602 | 0.3883 | 0.2148 | 3.5969 |
| ICT [44] | 2.7772 | 3.8382 | 0.4995 | 0.3328 | 4.2270 | 2.6783 | 3.4850 | 0.4950 | 0.3220 | 4.0722 | 2.3617 | 4.1360 | 0.4800 | 0.2820 | 3.8584 |
| MISF [27] | 2.3211 | 3.3633 | 0.3428 | 0.2349 | 3.8630 | 2.3384 | 3.1250 | 0.3779 | 0.2354 | 3.8342 | 2.1902 | **3.7318** | 0.3770 | 0.2132 | **3.5702** |
| VQFR [15] | 2.3838 | 3.4023 | 0.3800 | 0.2481 | 3.9027 | 2.4199 | 3.0511 | 0.3982 | 0.2503 | 3.8199 | 2.2896 | 3.8031 | 0.4248 | 0.2316 | 3.7070 |
| Ours | **2.1262** | **3.3405** | **0.3314** | **0.2278** | **3.7988** | **2.0899** | **2.9035** | **0.3523** | **0.2184** | **3.6453** | 2.0796 | 3.7534 | **0.3768** | **0.2122** | 3.6084 |

Table 3: Quantitative comparison for face fidelity. We use several face recognition networks to calulate identity distance between the generated face and ground truth face, which are used to measure face fidelity. Our method performs better in most metrics.

VoxCeleb with the same identity called VoxCeleb-ID. In addition, we collected two high-quality talking video datasets: FaceForensics++ [39], and HDTF[51]. They are collected from YouTube with a resolution of 720p or 1080p, most of which are clear front-face talking videos. We collect 761 and 358 videos from the URL provided by two datasets, then recognize and crop faces from video frames.

For face images, we align them along the eye area and resize all faces to 256 x 256. For the audio recordings, we crop all audio recordings into 6 seconds segments. If the audio length is not long enough, we repeat the audio to make it at least 6 seconds. The audio sampling rate is 16KHz, and the channel number is one. Following[48], we remove silence regions of each segment with a voice activity detector and extract a log mel-spectrogram using a Hann window of 25mm, 10ms hop, and 1024 FFT frequency bands. Finally, we get a 64x1000 dimensional vector for each audio segment.

Since the number of audio recordings is much more than faces in VoxeCeleb-ID, we randomly sample at most 20 audio segments for each identity. In FaceForensics and HDTF, we randomly sample 10 faces as they are similar from different video frames. Moreover, since our method does not focus on audio content, we ignore the lips change in FaceForensics++ and HDTF and manually select the standard faces which are frontal and lips change-free. For each standard face, We calculate a VGG-Face feature from a ResNet-50 pre-trained in the VGGFace2 [2] dataset as the face identity label that is used to calculate identity embedding loss. After all that, we got three pre-processed high-quality audio face-paired datasets, shown in Table 1.

**Model Pretraining.** We follow the idea from [48] for reconstructing faces from audio. However, since their method is trained in a low-resolution ($64 \times 64$) audio-face dataset, they can not generate high-quality faces. Therefore, we only borrow part of its parameters and add several upsampling layers to fit our high-resolution

| Methods | Acc@1 | Acc@5 | Acc@8 | Acc@10 |
|---------|-------|-------|-------|--------|
| CSA [29] | 17.37 | 34.21 | 41.05 | 45.26 |
| RFR [26] | 17.89 | 31.58 | 43.16 | 46.84 |
| MISF [27] | 17.37 | **38.95** | 42.11 | 46.84 |
| VQFR [15] | 14.21 | 31.58 | 39.47 | 40.53 |
| Ours | **20.53** | 35.79 | **46.84** | **49.47** |

Table 4: Face retrieval performance. We measure retrieval performance by accuracy at K (Acc@K, in %), which indicates the chance of retrieving the same person's faces within the top-K results while using the reconstruction faces of different methods.

dataset ($256 \times 256$). We train the audio face decoder with faces and audio segments in FaceForensics++ and HDTF. During training, we randomly select one face and one audio segment from the sample identity. The maximum training iteration is 100k.

**Training.** Since our method does not focus on the audio content and lips change, we train and evaluate our method with standard faces in three datasets. Given a face, we randomly select an audio segment and a face identity embedding of the same person. During training, we chose a rectangle mask with the size of $128 \times 128$ and set the learning rate as 2e-4 and batch size as 4. We chose ADAM as the optimizer with $\beta_1 = 0.5$, $\beta_2 = 0.999$ and train all the networks on an NVIDIA RTX 3090. More details of our method can be found in the supplementary material.

### 4.2 Results and Comparison

We evaluate our method and recent face inpainting methods with the standard faces on three datasets. PSNR, SSIM, LPIPS, L1 distance are used in our experiments to measure the pixel-level similarity between our result and ground truth face, and we also calculate a Landmark distance conducted by dlib to measure the structural similarity of faces. In order to evaluate the face fidelity, we

employ HOG, VGGFace[36], SphereFace [31], CosFace [46] and Ar-cFace [6] to measure the identity distance. The smaller the identity distance, the higher the face fidelity.

**Comparison with state-of-the-arts.** We compare our method with five recent face inpainting methods, CSA [29], RFR [26], ICT [44], MISF [27], and VQFR [15], in which ICT focus on pluralistic image completion, and VQFR focuses on blind face restoration. All methods are retrained in the same dataset for a fair comparison. We report the pixel and structural similarity comparison results in Table 2. Our method performs better than other methods in most metrics for all three datasets demonstrating its strength in generating high-quality faces. For identity preservation, we show quantitative results in Table 3. In FaceForensics++ and HDTF, our method outperforms all methods in all metrics and is slightly worse than MISF [27] in VoxCeleb-ID, which indicates the advantages of our approach in generating high-quality faces and face identity preservation.

We also show some qualitative results in Figure 5. Previous methods may generate face contents with mistaken identities when faces are missing in large areas. For example, in the first row of Figure 5, when the input face is mostly masked and the remaining area does not provide an identity reference, previous methods incorrectly generates a male face. In contrast, in our method, audio face learns an identity prior from sound, reconstructs a female face, and correctly guides our face decoder to generate identity-consistent female face. In addition, we show some results under three different sizes of irregular masks in Figure 6 to verify the universality of our method.

**Face Retrieval.** To verify that our method can generate results which are more closely related to the original facial features, we measure retrieval performance by accuracy at K (Acc@K, in %), which indicates the chance of retrieving the same person's faces within the top-K results while using the reconstruction faces of different methods as input. We used the test set of VoxCeleb-ID for this experiment, with 190 face images of different identities for retrieval and others for gallery. We query the face images by comparing the Euclidean distance of ArcFace face features between the reconstruction faces of different methods and the faces in the gallery. Table 4 shows the face retrieval performance. The experimental results show that our method can preserve the identity of faces more adequately.

## 4.3 Identity Swapping with Audio.

Although our method focuses on audio-driven identity preservation, it can also perform identity swapping with different reference audios. Since our face decoder is influenced by the audio embedding and intermediate features in the audio decoder, we can get a face with different identities if we change the input audio. We show the visualization results in Figure 7. For the same input face, if we input a young woman's voice, our method will generate a face with female features. On the other hand, if we change to the voice of a young man, the final result will be more like a male face. We also show some pluralistic results of ICT [44], which generates different faces without control. In contrast, our method can explicitly change the identity through audio and perform an audio-guided controllable face inpainting.

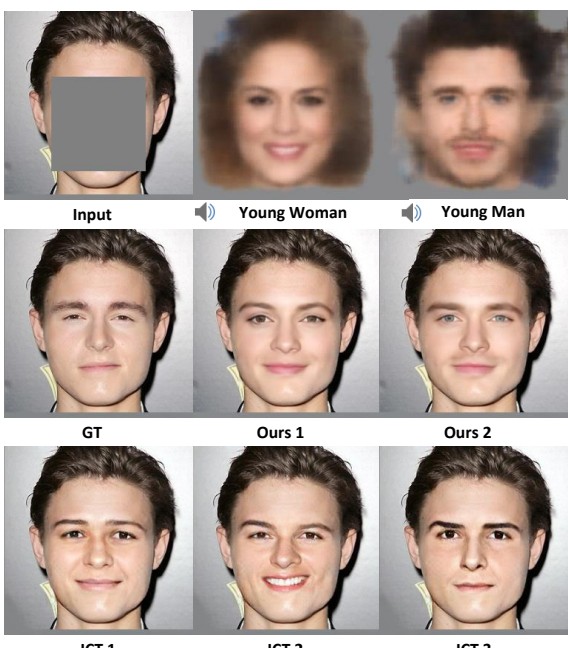

**Figure 7: Our method support the audio-control face inpainting. Given the audio of a young woman or a man, our method will generate a face with female or male features, respectively.**

## 4.4 Ablation Study

We conduct careful ablation study on VoxCeleb-ID dataset. Our baseline builds on a UNet. To validate the effectiveness of each component in our method, we add them in baseline one by one to observe how the result changes.

Quantitative and qualitative results are shown in Table 5 and Figure 8. When we only employ one audio embedding network to exploit the high-dimensional identity, the identity distance decreases while PSNR decreases. The reason is that a single decoder is challenging to reconstruct high-quality faces from the identity code, leading to face distortion and blur, shown in the third column in Figure 8. Since the features from the face decoder and audio face decoder are not pixel-aligned, directly concatenating two features will hurt the performance in all aspects. Although our AVFF module causes a decrease in PSNR and SSIM, it further reduces the identity distance compared to audio embedding, proving its effectiveness in identity preservation. We attribute the decline in PSNR and SSIM to the fact that the intermediate features of the audio decoder contain not only identity information but also other noise. We solve this problem by introducing identity consistency loss. The identity embedding loss constrains the space of the completed identity embedding with ground truth identity, which improves image quality and fidelity.

To demonstrate the role of audio for identity preservation, we extract the identity embeddings from the images generated by baseline model and our method, and visualize thems by t-SNE [43], shown in Figure 9. Our method achieves better clustering than

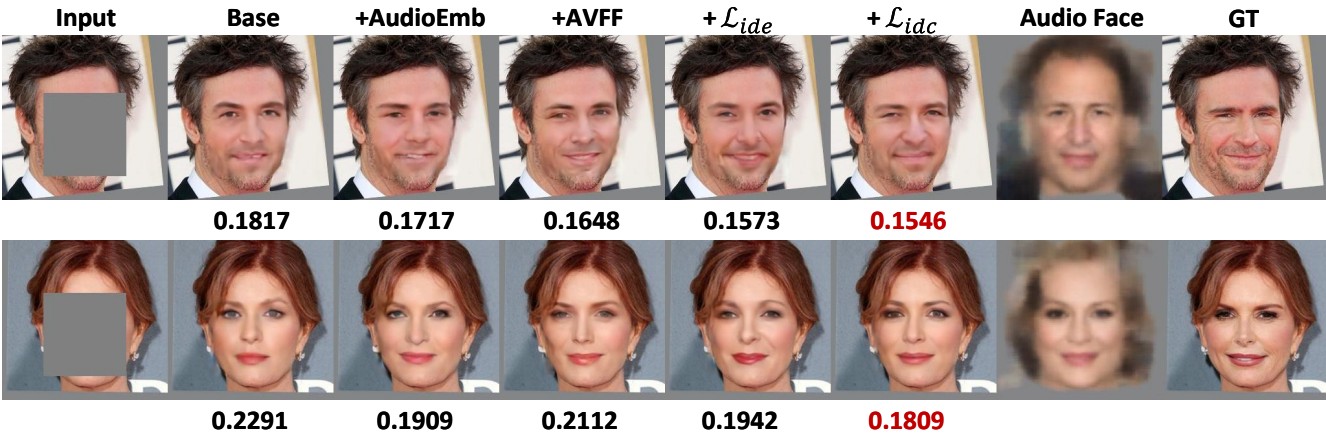

| Input | Base | +AudioEmb | +AVFF | $+\mathcal{L}_{ide}$ | $+\mathcal{L}_{idc}$ | Audio Face | GT |
|-------|------|-----------|-------|----------|----------|-----------|-----|
|  | 0.1817 | 0.1717 | 0.1648 | 0.1573 | **0.1546** |  |  |
|  | 0.2291 | 0.1909 | 0.2112 | 0.1942 | **0.1809** |  |  |

**Figure 8: Quantitative results of ablation study. CosFace[46] distance are blow the image. Our full method achieves a high-fidelity face.**

| Methods | VoxCeleb-ID | | | | |
|---------|------|------|------|------|------|
|  | ↑ PSNR | ↑ SSIM | ↓ LPIPS | ↓ CosFace | ↓ ArcFace |
| Base | 25.9169 | 0.8942 | 0.0592 | 0.2176 | 3.6722 |
| Base + AudioEmb | 25.8960 | 0.8947 | 0.0574 | 0.2164 | 3.6477 |
| Base + AudioDec + Concat | 25.8909 | 0.8944 | 0.0577 | 0.2167 | 3.6461 |
| Base + AudioDec + AVFF | 25.8292 | 0.8938 | 0.0548 | 0.2156 | 3.6446 |
| Base + AudioDec + AVFF + $\mathcal{L}_{ie}$ | 25.9368 | 0.8956 | 0.0568 | 0.2137 | 3.6291 |
| Base + AudioDec + AVFF + $\mathcal{L}_{ie}$ + $\mathcal{L}_{ic}$ | **25.9633** | **0.8963** | **0.0544** | **0.2122** | **3.6084** |

**Table 5: Quantitative results of ablation study on VoxCeleb-ID. Each of the components we propose is effective in reducing identity distance and improving fidelity.**

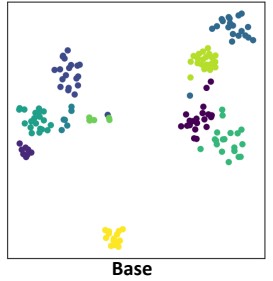 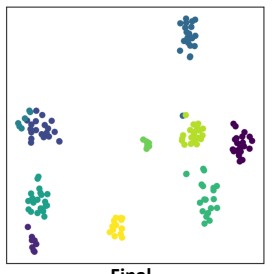

**Figure 9: t-SNE [43] visualization of identity embeddings from some inpainted faces by baseline and our method (denoted as Final). Our method achieves better aggregation degree.**

baseline indicating that audio is useful to provide discriminative identity information.

## 5 DISCUSSION

**Potential Social Impact.** Our method can achieve controlled face inpainting through changing audio. On the one hand, the encoding of facial identity may cause privacy leakage. On the other hand, face swapping produces fake face images, which may deceive the face recognition systems. Thus, this technique should be used with caution.

**Limitation.** Our method realizes the generation of high fidelity faces by interacting with audio features. The results are affected by the quality of audio faces. Although our method can produce real-looking audio faces, the facial details need to be enhanced. Due to the limitation of identity number in our datasets, our model can only express several simple attributes, such as age and gender. What audio can bring to face inpainting needs more exploration.

## 6 CONCLUSION

This paper first verifies the critical role of audio in face inpainting. We propose an audio-driven high-fidelity face inpainting method. It captures implicit and explicit identity representations from audio and learns deterministic priors from input faces via a dual-stream network. We design an audio-visual feature fusion module that can effectively integrate multi-scale deep features of cross-modal data. We also introduce two identity losses for preserving face identity. Experiments show that voice can help generate face structure and identity prior, and our method can generate high-fidelity faces with audio guidance.

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
