# OpenReview forum: "Audio-Driven Identity Manipulation for Face Inpainting"
_acmmm.org/ACMMM/2024/Conference — MM2024 Poster_

### Official Review · Reviewer_w5d8 · 2024-05-02

**Rating:** 4
**Confidence:** 4

**Summary:**

This paper introduces audio signal into the face inpainting task and recover the target face whose identity corresponds to the property of the audio signal. Also, the audio-visual feature fusion (AVFF) module is proposed to fuse multi-scale features from the face and audio decoder. The identity embedding loss and the identity consistency loss are designed to train the module.

**Strengths:**

This paper is the first to introduce audio guidance into the face inpainting task and can serve as an inspiration for future related works. Also, the writing is good and the paper is quite easy to follow.

The dataset preparation is thorough and experimental results are comprehensive enough to support the proposed method.

**Limitations:**

The motivation of the network design in Audio-Visual Feature Fusion Module is a little unclear. It seems meaningless to add the identity feature map and what is the design principle? What do the add branch and concat branch stand for?

How to choose the mask size and location in the face image? Does it impact the performance?

From the qualitative results, the center area of the generated faces is somehow faint compared to other methods. Why getting leading scores on PSNR and SSIM metrics? Is it related to the input visible areas of the face? Adding metric like CPBD could be more convincing.

The experimental setting is unfair. Additional information like audio signal is provided to the propose method during the generation process, while other compared methods get information no more than prompt word like male or female. How to explain the experimental setting?

Some statements are ambiguous. a) What does the word manipulation in refer to? The proposed model seems to recover the face identity under the guidance of audio clues. The terms inpainting and manipulation seem a little contradictory. b) I_i is not marked in Figure 3.

**Suitability:**

3

---

### Official Review · Reviewer_PX86 · 2024-05-05

**Rating:** 3
**Confidence:** 3

**Summary:**

This paper proposes a method to infer visual identity information from a speaker’s voice and utilizing the extracted identity information for face inpainting and face swapping. More specifically, an audio encoder is utilized to extract the identity information. These identity information is then fused into the cropped face by a face decoder to serve as an identity guidance to reconstruct the face.

**Strengths:**

①The task is interesting and the connection between human voice and visual appearance has the potential for a wide range of application.

②The paper is well organized and easy to follow.

**Limitations:**

### Weakness and Question

① My primary concern lies in the correspondence between the voice identity and face identity. While this paper posits that this correspondence is a deterministic one-to-one relationship, I contend that it should be a more ambiguous mapping (a many-to-many relationship). For instance, both of the following scenarios are plausible: individuals with similar appearances may exhibit entirely different vocal characteristics, and conversely, people with similar voices may possess completely distinct facial features. Therefore, employing probabilistic generative models, such as Variational Autoencoders (VAEs) or diffusion models, may better capture the ambiguous mapping. Deterministic models like GANs may lead to a neutral face (the reconstrcted audio face $x_a$ in this paper).

② When preparing masks for facial regions, the center crop masks are relatively coarse and may result in models learning not only facial identity information but also some redundant background information. Utilizing finer-grained masks, such as the results obtained by facial segmentation or anonymization, would be more effective.

③ The quantitative ablation experiments in both the main text (table 5) and supplementary materials (table 1) indicate that many of the proposed modules do not make significant contributions, which raises doubts about the necessity of the modules proposed in this paper.

④ Some further experiments may make the proposed method more convincing. For instance, it would be informative to explore the outcomes of guiding the same masked face:
+ with/without audio
+ audios with different identity but the same content
+ audios with the same identity but different contents

Similarly, investigating the effects of using the same audio to guide different masked faces would provide valuable insights.

**Suitability:**

3

---

### Official Review · Reviewer_LrBh · 2024-05-20

**Rating:** 5
**Confidence:** 3

**Summary:**

The paper presents an innovative approach to face inpainting that integrates audio cues to facilitate identity manipulation. The core idea is that a person's voice contains distinct identity markers, such as age and gender, which can enhance face inpainting tasks. The authors propose a dual-stream network architecture that includes a face branch for deterministic identity extraction and an audio branch for heuristic identity priors. The system generates a unified identity embedding through a multi-layer perceptron (MLP) and introduces an audio-face generator to fuse multiscale features into the face inpainting network via an audio-visual feature fusion (AVFF) module. Experiments demonstrate the method's effectiveness, particularly in identity preservation.

**Strengths:**

Novelty: This paper unprecedentedly introduces the audio modality into the face inpainting task, representing a significant departure from traditional approaches that rely solely on visual information. By leveraging the unique identity cues embedded in a person's voice, such as age, gender, and emotional state, the authors propose a dual-stream network architecture that integrates audio analysis with facial feature reconstruction. This cross-modal strategy not only enhances the fidelity of face inpainting but also introduces a new dimension for identity manipulation, allowing for both identity preservation and swapping.
Application: The paper illustrates its strength in identity preservation, which aligns well with the identity swapping task, suggesting broad applications for the proposed method. The ability to manipulate facial identity while maintaining high fidelity in image reconstruction opens up numerous possibilities across various fields.

**Limitations:**

This article discusses the task of facial reconstruction using audio information [47] and proposes the novel application of audio information for facial inpainting tasks. What are the unique challenges and difficulties in utilizing audio information, especially for the task of facial inpainting? The authors mention the potential future applications of this work for face recognition and facial attribute classification in the introduction (lines 106-109). What considerations are behind this suggestion? Overall, this paper is a pioneering work in the task of facial repair using audio information, but it lacks a more thorough comparison with other facial tasks that also utilize audio information. In particular, the paper does not sufficiently discuss the potential value of using audio information for applications such as face recognition and facial attribute classification.

**Suitability:**

2

---

### Official Review · Reviewer_DiGe · 2024-05-24

**Rating:** 4
**Confidence:** 2

**Summary:**

Introduce the concept of audio face to improve the face inpainting task. The proposed the AVFF module improves the identity reservation of the overall generation model.

**Strengths:**

1. good visual effect  of the generated image
2. maintain identity consistency better than the previous methods

**Limitations:**

The audio contains various types of information, such as rhythm, tone, voice, and speaking content. A more fine-grained disentanglement design is needed to improve the identity extraction ability. Relying solely on a speaker recognition model might introduce bias based on the training dataset used for the speaker recognition task.

**Suitability:**

3

---

### Meta-Review · Program_Chairs · 2024-07-14

**Recommendation:** Accept (Poster)
**Confidence:** 4

**Metareview:**

this paper investigates the face generation problem with multi-modal conditioning. specifically, it argues that the audio modality can provide additional information like gender and age other than the visual surroundings. a dual stream network is proposed for this task and is verified on face inpainting benchmarks.

initially, the paper received ratings of BA, WA, BR, BA. reviewers had mixed feelings about it. introducing the audio modality is well recognized by the reviewers, and similarly appreciated are the visual results, the potential application, and overall writing. there are concerns over the biasing the system through the usage of a voice recognition system, voice-to-face being a one-to-many mapping and not sufficiently modelled in this work, lack of discussion and comparison with related work, the face cropping out and the experimental settings, and some clarity issues. during the rebuttal, the authors provided feedback for these concerns. while there remains some un-resolved issues regarding the mask in the training and the ablations, the reviewers overall post postive feedbacks.

since the AC was unable to finish the meta review in time, the PC stepped in and carefully went over all the reviews and rebuttal. after consideration, the PC recommends Accept. with that said, the authors are strongly encouraged to modify the manuscript based on the rebuttal and further clarify issues regarding the the mask in the training and the ablations.